

# Flow intermittence prediction using a hybrid hydrological modelling approach: influence of observed intermittence data on the training of a random forest model

Louise Mimeau[1], Annika Künne[2], Flora Branger[1], Sven Kralisch[2], Alexandre Devers[1], and Jean-Philippe Vidal[1]

[1]INRAE, UR RiverLy, Centre de Lyon-Villeurbanne, 5 rue de la Doua CS 20244, 69625 Villeurbanne, France
[2]Geographic Information Science Group, Institute of Geography, Friedrich Schiller University Jena, Löbdergraben 32, 07743 Jena, Germany

**Correspondence:** Louise Mimeau (louise.mimeau@inrae.fr)

**Abstract.** Rivers are rich in biodiversity and act as ecological corridors for plant and animal species. With climate change and increasing anthropogenic water demand, more frequent and prolonged periods of drying in river systems are expected, endangering biodiversity and river ecosystems. However, understanding and predicting the hydrological mechanisms that control periodic drying and rewetting in rivers is challenging due to a lack of studies and hydrological observations, particularly in non-perennial rivers. Within the framework of the Horizon 2020 DRYvER (Drying River Networks and Climate Change) project, a hydrological modelling study of flow intermittence in rivers is being carried out in 3 European catchments (Spain, Finland, France) characterized by different climate, geology and anthropogenic use. The objective of this study is to represent the spatio-temporal dynamics of flow intermittence at the reach level in meso-scaled river networks (between 120 km$^2$ and 350 km$^2$). The daily and spatially distributed flow condition (flowing or dry) is predicted using the J2000 distributed hydrological model coupled with a Random Forest classification model. Observed flow condition data from different sources (water level measurements, photo traps, water temperature measurements, citizen science applications) are used to build the predictive model. This study aims to evaluate the impact of the observed flow condition dataset (sample size, spatial and temporal representativity) on the performance of the predictive model. Results show that the hybrid modelling approach developed in this study allows to accurately predict the spatio-temporal patterns of drying in the 3 catchments. This study shows the value of combining different sources of observed flow condition data to reduce the uncertainty in predicting flow intermittence.

## 1 Introduction

River systems are an essential link in terrestrial biodiversity. They are home to many animal and plant species within the riverbed and in the riparian zone (Leigh and Datry, 2017). They also serve as ecological corridors by providing a connection between upstream and downstream for mobile species and by transporting nutrients and sediments necessary for the survival of species located downstream (Deiner et al., 2016). These ecological corridors can be disrupted when river beds dry up. By impacting the hydrological cycle and increasing the risk of drought (Gudmundsson and Seneviratne, 2016; Tramblay et al.,



2021), climate change threatens river biodiversity (Bond et al., 2008). Prolonged drying and shifting of river sections from perennial to intermittent flow can endanger ecosystems and the access to water resources useful to our society (Steward et al., 2012; De Girolamo et al., 2017; Tonkin et al., 2019).

Although they represent a large proportion of terrestrial rivers (Messager et al., 2021), intermittent rivers are still poorly known (Acuña et al., 2014; Meerveld et al., 2020; Fovet et al., 2021) and their study in hydrology is relatively recent. Modelling the hydrological functioning of drying river networks (DRNs) can help understanding the impact of drying on ecosystems and to predict the evolution of the drying spells and possible tipping points in flow regimes under climate projections.

Studies have already looked at modelling intermittent rivers with a physical hydrological model (Jaeger et al., 2014; Tzo-
raki et al., 2016; Llanos-Paez et al., 2023). One major difficulty in modelling flow intermittence is that hydrological models have difficulties to simulate no flows (Shanafield et al., 2021). First there is a numerical challenge: the flow routing scheme implemented in the models to propagate the streamflow across the river networks cannot represent sudden transitions from wet to dry. Second, the origins of intermittent rivers are multiple (disconnection between the river and the water table, drying up following a long period without precipitation, infiltration from the river bed into a fault or a karstic subsoil, drying up
following anthropic withdrawals, etc.) (Datry et al., 2016) and sometimes very local. Representing all these processes in the models is thus complex and requires a large amount of data. A more common modelling approach to model intermittent rivers is the use of artificial neural networks (ANN) (Daliakopoulos and Tsanis, 2016; Beaufort et al., 2019) and random forest (RF) (González-Ferreras and Barquín, 2017; Beaufort et al., 2019; Belemtougri, 2022; Jaeger et al., 2023) models. These models are easier to implement, do not require a priori knowledge of the origins of drying, and showed good performances to predict
the spatial distribution of flow regimes (perennial or intermittent) in the river networks. The covariates used to predict the rivers flow regimes are usually the streams physical characteristics (width, length, slope, geological context, etc) and climatic variables such as precipitation, temperature and evapotranspiration. The prediction of the spatial and temporal dynamics of drying in intermittent river systems, requires providing the RF models with additional covariates on the spatialized hydrological conditions along the river systems at a significantly fine time step and fine spatial resolution. This can be achieved using
spatially distributed hydrological models at a daily or smaller time step.

Another a challenge in the study of intermittent river networks is to collect observed data of flow intermittence to train or validate the models. Studies of river intermittency on a large scale mainly use gauging station data (Belemtougri, 2022; Messager et al., 2021; Tramblay et al., 2021; Beaufort et al., 2019; Reynolds et al., 2015). Gauging station data are easy to retrieve and analyse and have the advantage of providing data at a regular time step over long periods. But the stations are
mainly located on rivers with perennial flow (Eng et al., 2016; Meerveld et al., 2020) and their spatial distribution is not dense enough do understand the flow intermittence patterns along river networks. On the contrary, studies focusing on smaller catchments use data from field campaigns (Jaeger et al., 2023; Llanos-Paez et al., 2023; Sefton et al., 2019), which allow the collection of data at regular time steps with a denser network of observations. But field campaigns can be costly and time consuming, and usually cover short periods of time (several weeks or month), with a risk of over-representing drying events
when the campaign is focused on the summer season.





The objective of the study is to present a hybrid modelling approach to simulate spatio-temporal patterns of drying in the river networks. To do so, we developed a flow intermittence model by coupling a distributed hydrological model (JAMS-J2000) with a Random Forest classification model. The models are applied in 3 European DRNs from the DRYvER project (Datry et al., 2021) located in Spain, France and Finland to evaluate the ability of the models to predict the drying patterns in contrasted
climate, hydrological, geological and anthropogenic contexts.

In this study, we developed a flow intermittence model by coupling a distributed hydrological model (JAMS-J2000) with a Random Forest classification model to simulate spatio-temporal patterns of drying in the river networks. The models are applied in 3 European DRNs located in Spain, France and Finland to evaluate the ability of the models to predict the drying patterns in contrasted climate, hydrological, geological and anthropogenic contexts.

This study also investigates the different types of observed flow state data available to drive the RF model (gauging stations, field campaigns, crowdsourced data, remote sensing, expertise), their ability to represent the actual drying patterns in the DRNs, and how they can be combined to improve the modelling of flow intermittence.

## 2   Method

### 2.1   Study area and data

#### 2.1.1   Focal DRNs

This study focuses on 3 meso-scale DRNs located in Spain, France and Finland (Table 1, Fig. 1) that are part of the DRYvER project on drying rivers and climate change (Datry et al., 2021). The 3 catchments have similar areas ranging between 200 and 350 km$^2$ and are characterized by different climates and flow intermittence patterns.

The Genal catchment, located in South of Spain, is characterized by a dry and warm climate and scarce natural vegetation.
Long periods of drying are observed in the smaller reaches. The main Genal river is known to be perennial except in the downstream part of the catchment where the Genal river dries up in the summer season due to water abstraction for irrigation.

On the opposite, the Lepsämänjoki catchment in Finland is characterized by a wetter and colder climate. Flow intermittence is only observed in the smallest reaches but seems to have intensified in recent years due to climate change.

The Albarine in France is characterized by a more temperate climate. Flow intermittence is particularly observed in the
upstream and downstream parts of the catchment. Drying is mainly due to the seepage of the Albarine river in the soil at geological discontinuities.



| Name | Country | Area [km²] | Outlet lat. [°] | Outlet lon. [°] | Range of elevations [m.a.s.l] |
|---|---|---|---|---|---|
| Albarine | France | 354 | 45.906 | 5.234 | 212 - 1497 |
| Genal | Spain | 343 | 36.318 | -5.312 | 3 - 1718 |
| Lepsämänjoki | Finland | 208 | 60.238 | 24.984 | 8 - 145 |

**Table 1.** characteristics of the studied catchments

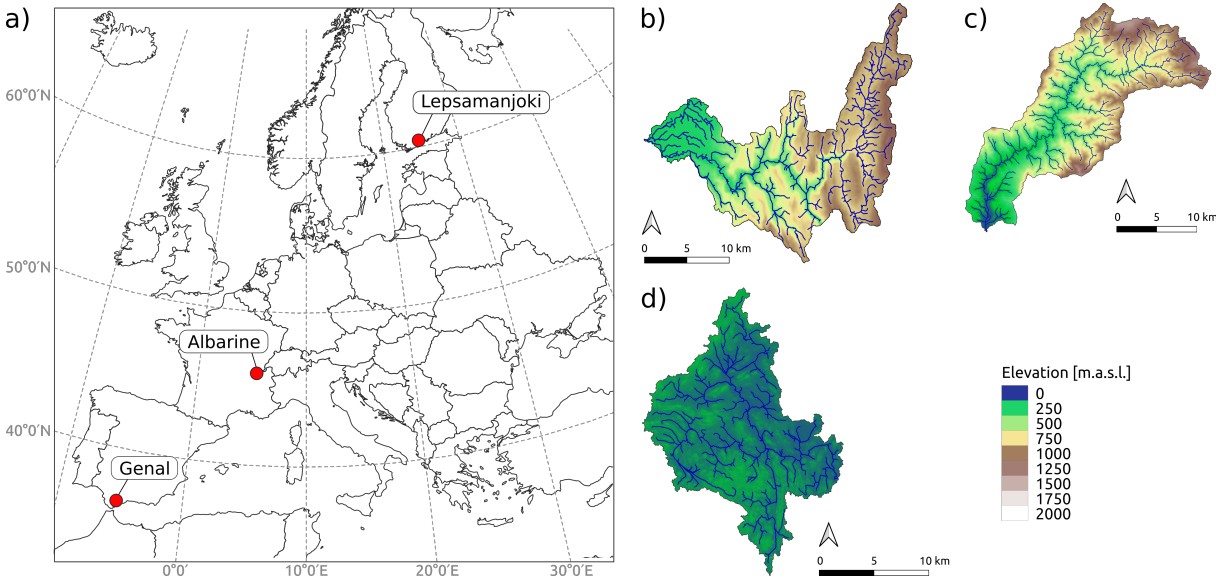

**Figure 1.** a) Location of the 3 studied DRNs. b,c,d) River networks and elevations of the Albarine, Genal, and Lepsämänjoki DRNs.

### 2.1.2 Spatial data

Topography, soil types, landuse and hydrogeology information is needed for the spatially distributed hydrological modelling. The following data sources were used:

– Topography: EU-DEM v1.1 (Copernicus, 2016) with a 25 m resolution for the Albarine DRN, the Andalucía DEM (Portal Ambiental de Andalucía. 2010) with a 10 m resolution for the Genal DRN, and the 10 m DEM Finland (National Land Survey of Finland) for the Lepsämänjoki DRN.

– Soil: Soil classes were used from the European Soil Database v2.0 (European Commission; Panagos et al. (2012)). Physical parameters were also used from the European Soil Database v2.0 (field capacity, saturated water content, depth
to rock). In the Genal DRN (Spain) texture and bulk density data was used from soil profiles (Llorente et al., 2018) for the calculation of parameters using pedotransfer functions (Ad-Hoc-AG, 2005; Baxter, 2007).



- Land use: Corine Land Cover (CLC) 2012, Version 2020-20u1 Level 3 (44 classes) was used (Copernicus Land Monitoring Service 2020) to establish the LULC classes. Parameters, such as albedo, crop coefficients, LAI, root depth, and impervious fraction area, were adapted to local conditions from different sources (Allen et al., 1998; Krause, 2001; Kralisch and Krause, 2006; Neitsch et al., 2011; Faroux et al., 2013)(Allen et al. 1998, Krause 2001, Krause et al. 2006; Ludwig and Bremicker 2006; Neitsch et al. 2011, Faroux et al. 2013).

- Hydrogeology: IHME1500 – International Hydrogeological Map of Europe (aquifer and litholgy layers)(Duscher et al., 2015) was used to establish the classes for all DRNs.

### 2.1.3 Climate data

The ERA5-land reanalysis (Muñoz-Sabater et al., 2021) was used to as climate forcing data for the hydrological modelling. The following hourly ERA5-land climate variables were used to compute the reference evapotranspiration using the Penman-Monteith equation (Allen et al., 1998): 2 m air temperature (°C), 2 m dew point temperature (°C), 2 m relative humidity (%), 10 m u and v wind speed components (m/s, incoming solar radiation (W/m2), incoming thermal radiation (W/m2), and surface pressure (Pa). Hourly ERA5-land precipitation, air temperature and computed reference evapotranspiration were then aggregated at daily time step to be used as climate forcing data in the hydrological model.

### 2.1.4 Flow state and and discharge data

In order to validate the models ability of simulating flow intermittence at the reach level multiple data sources of flow observations were used:

- hydrological stations: (i) discharge daily time series from gauging stations (http://leutra.geogr.uni-jena.de/DRYvER). The streams are considered as dry if the measured discharge is equal to 0 m3/s, and flowing otherwise. (ii) ONDE network (Observatoire National des Etiages, https://onde.eaufrance.fr): this French network of hydrological stations was specifically developed to monitor intermittent rivers and gives a monthly qualitative information about the state of flow (visible flow, non-visible flow, dry).

- crowdsourced data from smartphone applications: DRYRivERS (https://www.dryver.eu/app) and CrowdWater (https://crowdwater.ch/en/data/).

- measurements from field campaigns for the DRYvER project : phototraps installed along the river networks took daily pictures from 2018-11-07 to 2022-04-30 in the Albarine DRN and from 2021-06-17 to 2021-09-26 in the Lepsämänjoki DRN.

- observations on Google Earth images: the state of flow of the reaches was observed on the images for several dates between 2010 and 2022. The observation with Google Earth images was only possible in the Genal DRN which has a scarce vegetation.



- expertise of local DRYvER project partners : some members of the DRYvER project have been studying these DRNs
  for several years and have a deep understanding or their hydrological behaviours. Their expertise was used to identify
  reaches characterised by a perennial flow. Reaches are assumed to be flowing every day during the field campaign period.

These data sources are available either as disconnected points in time and space (Fig. 2), recurrent observations at the sampling
sites or time series of daily data over periods ranging from a few months to several years.

In case there are several flow state observations on the same day in a reach, only one observation is kept to train the RF
model. First a filter is applied to prioritize data from direct observations (e.g. ONDE stations, crowdsourced data, phototraps,
Google Earth) and remove data from indirect measurements (gauging stations). If after this selection, there are still more than
one observation per reach and per day, only one observation with the predominantly observed flow state is (flowing or dry) is
kept.

A detailed analysis of the flow state observations and their ability to represent the drying in the river networks is presented
in the Results section (Table 5 and Fig. 5).

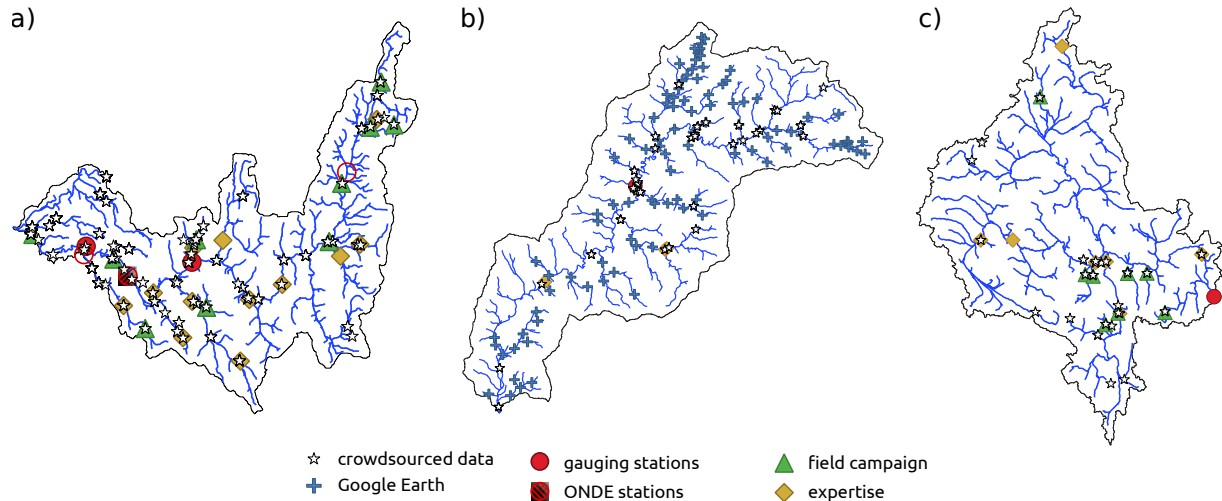

**Figure 2.** Observed state of flow data in the a) Albarine, b) Genal, c) Lepsämänjoki DRNs.

## 2.2 Flow intermittence model

In order to simulate flow intermittence, a spatially distributed process-oriented hydrological model (JAMS-J2000) was imple-
mented on mesoscaled DRNs (detailed description of the model in Section 2.2.1). Once calibrated and validated, the JAMS-
J2000 hydrological model enables to simulate daily streamflow time series in each reach of the river network.

Then, the deterministic hydrological model was coupled with a stochastic model, using the model outputs and physical
information to train a Random Forest (RF) classification model with some flow-state observations. The outputs of the RF
model enables to predict the daily flow state (flowing or dry) in each reach of the DRN, and thus predict the spatio-temporal
patterns of flow intermittence.





The modeling method to simulate flow intermittence is summarized in Figure 3 and is described in detail in the following sections.

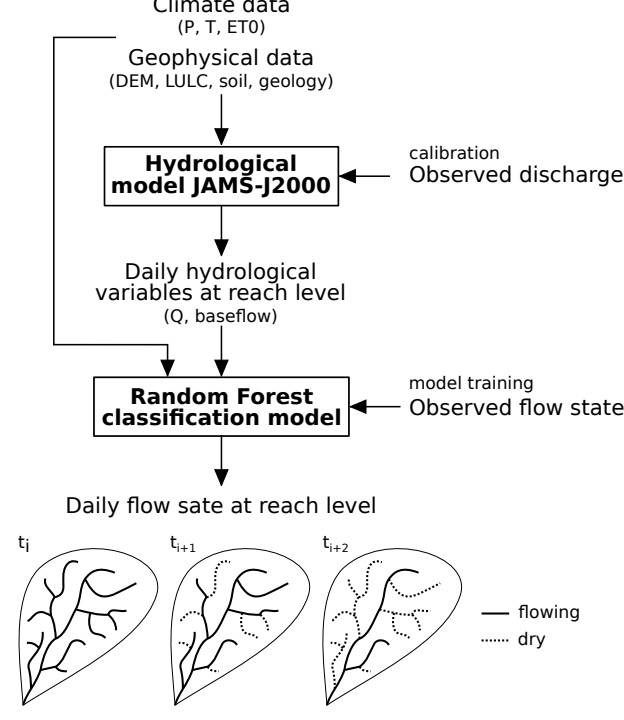

**Figure 3.** Modelling approach to simulate flow intermittence in river networks by coupling a distributed hydrological model to a Random Forest classification model

### 2.2.1 JAMS-J2000 hydrological model

The process-oriented JAMS-J2000 hydrological model (Kralisch and Krause, 2006) is used to simulate spatially distributed hydrological variables in the DRNs. The catchment represented in JAMS-J2000 is discretized in Hydrological Response Units (HRU) From climate forcing data, JAMS-J2000 simulates evapotranspiration, snow processes, soil water balance and groundwater processes at the HRU level and computes lateral flow routing to account for surface, sub-surface, and groundwater flow from hillslopes into the stream and along stream segments until the outlet of the river network (Figure 4).

The J2000 river networks were generated from the flow directions and flow accumulations computed from the DEMs. Observed river networks were used to validate the generated river networks and make sure that the J2000 river networks corresponds to the observed river networks (see Fig. S1, S2, and S3 in supplementary material).

Some modifications from the standard J2000 hydrological model were made for this study using the evapotranspiration module from Branger et al. (2016) to compute potential evapotranspiration using the reference evapotranspiration and spatially 155   distributed crop coefficients. Besides, the adapted J2000 snow module by Gouttevin et al. (2017) was used.





Daily hydro-meteorological variables such as spatially distributed discharges and groundwater contributions, as well as evapotranspiration, snowmelt, soil saturation and groundwater saturation at the catchment scale are simulated with the JAMS/J2000 model from 01/10/2005 to 30/04/2022.

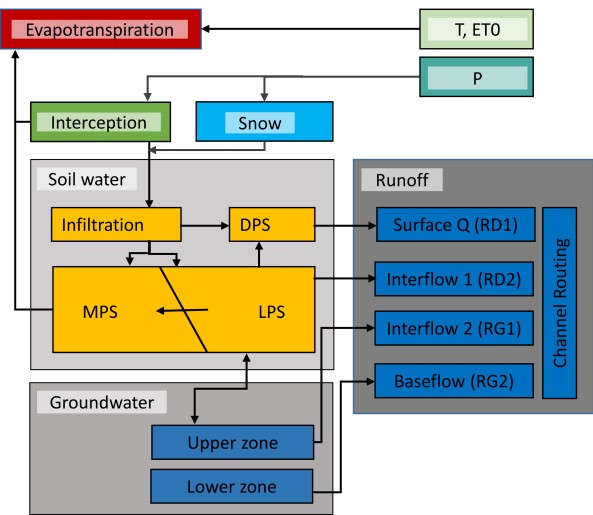

**Figure 4.** Schematic representation of the hydrological processes modeled in JAMS-J2000 at the HRU and reach level according to Krause (2001), Figure adapted from Watson et al. (2020)

### 2.2.2   Calibration of JAMS-J2000 model

This section only describes the general aspects of the method used to calibrate the JAMS-J2000 model. A full description of the calibration method as well as parameter values for each DRN are presented in the supplementary material (Tab. S1 and S2).

Calibration of the JAMS-J2000 parameters was performed on larger catchments (1500 to 3700 km$^2$) corresponding to the intermediate-scale basins studied in the DRYvER project (to bridge the gap between the DRN scale and the continental scale).

First, for the Albarine and Lepsämänjoki catchments, 4 lumped parameters for snow processes were calibrated to optimize

the simulated snow cover area. Then, 15 lumped parameters and 4 distributed parameters were calibrated in order to optimize the simulated discharges at the gauging stations. The Kling-Gupta-Efficiency (Gupta et al., 2009) was used to assess model performance, as well as different evaluation criteria focusing on low flows, such as the 10$^{th}$ percentile of the discharge. The calibration and validation periods for the 3 DRNs are presentented in Table 2.

| DRN | Initialization period | Calibration period | Validation period |
|---|---|---|---|
| Albarine | 1990-1995 | 1995-2009 | 2009-2020 |
| Genal | 1998-2001 | 2001-2004 | 2012-2018 |
| Lepsämänjoki | 2000-2005 | 2005-2014 | 2014-2020 |

**Table 2.** Calibration and validation periods. Hydrological years start on the 1$^{st}$ of October and end on the 30$^{th}$ of September.





Table 3 shows the performance of the JAMS-J2000 model to simulate the discharges at the locations of the gauging stations
in the 3 DRNs. KGE values for the calibration and validation periods show that the discharges are well simulated by the
hydrological model. The comparison between the simulated and observed 10th percentile of discharge also shows that JAMS-
J2000 gives good results for low flows. In the Albarine DRN, the Saint-Denis-en-Bugey station is located in the downstream
part of the river, which is intermittent due to the seepage of the Albarine river in the aquifer. This explains the poorer results
on this station as the seepage of the Albarine river is not represented in JAMS-J2000 model.

| DRN | Gauging stations | KGE calib. / valid. | Q10sim calib. / valid. [m3/s] | Q10obs calib. / valid. [m3/s] |
|---|---|---|---|---|
| Albarine | Saint-Rambert-en-Bugey | 0.76 / 0.79 | 0.94 / 0.50 | 0.74 / 0.51 |
| | Saint-Denis-en-Bugey | 0.55 / 0.69 | 0.86 / 0.45 | 0 / 0 |
| Genal | Jubrique | 0.75 / 0.76 | 0.01 / 0.02 | 0.11 / 0.05 |
| Lepsämänjoki | Lepsämänjoki | 0.74 / 0.81 | 0.56 / 0.40 | 0.56 / 0.33 |

**Table 3.** Validation of the JAMS-J2000 model. KGE values for the calibration and validation periods, and comparison between simulated
and observed 10th percentile of discharge during the calibration and validation periods.

### 2.2.3 Random Forest classification model

The results of the JAMS-J2000 hydrological models are used to train a Machine Learning model to predict the flow inter-
mittence at the reach level. The Random Forest (RF) classification and regression model (Breiman, 2001) is used to predict
the daily state of flow (dry or flowing) at the reach level. The RF model uses 20 covariates (based on Beaufort et al. (2019))
(Table 4):

– reach physical characteristics: drainage area, slope, type of landuse, type of soil, hydro-geological class around the
reaches);

– daily hydro-meteorological variables aggregated at the catchment scale: incoming liquid water, temperature and actual
evapotranspiration during the 10, 20 and 30 previous days as well as soil and groundwater saturation;

– spatially distributed hydrological variables simulated with JAMS-J2000: discharge and groundwater contribution (at $t_0$,
and average during the 10 previous days).




| Abbreviation | Description | Spatial / temporal distribution | Data source |
|---|---|---|---|
| drained_area | Drainage area of the reach | reach / uniform | Albarine: EU-DEM v1.1 Genal: Andalucia DEM Vantaanjoki : DEM Finland |
| slope | Slope of the reach | reach / uniform | Albarine: EU-DEM v1.1 Genal: Andalucia DEM Vantaanjoki : DEM Finland |
| landuse | Majority landuse class of HRUs crossed by the reach | reach / uniform | Corine Land Cover 2012 |
| soil | Majority soil class of HRUs crossed by the reach | reach / uniform | European Soil Database v2.0 |
| hgeo | Majority hydro-geological class of HRUs crossed by the reach | reach / uniform | IHME1500 |
| R10, R20, R30 | Sum of incoming liquid water (rainfall + snowmelt) during the 10, 20 and 30 previous days | catchment / daily | simulated with JAMS-J2000 |
| T10, T20, T30 | Mean air temperature during the 10, 20 and 30 previous day | catchment / daily | ERA5-land |
| ET10, ET20, ET30 | Sum of actual evapotranspiration during the 10, 20 and 30 previous days | catchment / daily | simulated with JAMS-J2000 |
| SoilSat | Mean saturation of the soil reservoirs | catchment / daily | simulated with JAMS-J2000 |
| GwSat | Mean saturation of the groundwater reservoirs | catchment / daily | simulated with JAMS-J2000 |
| Q | River discharge | reach / daily | simulated with JAMS-J2000 |
| Q10 | Mean river discharge during the 10 previous days | reach / daily | simulated with JAMS-J2000 |
| GW | Groundwater contribution to the river discharge | reach / daily | simulated with JAMS-J2000 |
| GW10 | Mean groundwater contribution to the river discharge during the 10 previous days | reach / daily | simulated with JAMS-J2000 |

**Table 4.** List of the covariates used in the RF model to predict the spatially distributed daily state of flow.

The RF models were implemented and calculated using the R package "ranger" (Wright et al., 2020).





For each DRN, the RF models are trained using flow state observations, and then used to extrapolate spatially and temporally the daily state of flow in each reach during the simulation period (01/10/2005 - 30/04/2022). To use the most of the observed flow state data (Section 2.1.4), the RF model is trained with all available data.

During the training phase of a RF model, a subset of variables is randomly selected at the node's splitting point in each random forest tree (Breiman, 2001). In this study, the RF is trained 20 times in order to take into account this structural uncertainty.

The ability of the RF model to represent flow intermittence is evaluated with two efficiency criterias: the probability of prediction of dry events (POD) and the false alarm ratio of dry events (FAR).

$$POD = \frac{a}{a+c}$$

$$FAR = \frac{b}{a+b}$$

with $a$ the number of dry observations correctly simulated by the model, $b$ the number of flowing observations that were simulated as dry, and $c$ the number of dry observations that were simulated as flowing.

**2.3 Sensitivity analysis of the RF model**

**2.3.1 Sensitivity to the size of the training sample**

First, the sensitivity of the RF model to the size of the training sample is tested by randomly selecting 75% of the flow state observations to train the RF model for each of the 20 runs. The RF model is then evaluated on the remaining 25%. For the 20 runs, the selection of the 75% of training data is based on a different random draw. This first test aims at evaluating the impact 205 of using a reduced training dataset on the prediction of flow intermittence. It also aims at evaluating the error of the RF model on a validation sample.

**2.3.2 Sensitivity to the type of flow state observed data**

As presented in Section 2.1.4, the collected flow state observation datasets used to train the RF model are heterogeneous in terms of spatial and temporal distributions of the observations, and representativity of different types of flow regimes. The 210 sensitivity of the RF model to each type of observed data (stations, field campaign, crowdsourced data, Google Earth, expertise) is evaluated by removing in turn each type of data from the training dataset, and then comparing the RF performances and the predicted flow intermittence patterns. The RF performances are evaluated on the whole dataset of flow state observations in order to compare the performance on the same validation dataset. The objective of this analysis is to assess the amount of useful information brought by each type of data.





### 2.3.3 Sensitivity to the geology data

The last test aims at analysing the sensitivity of the RF model to different degrees of accuracy of covariates. Here, we focus on the study case of the Albarine DRN, in which a main cause of intermittence is the infiltration of the riverbed in moraine deposits or karstic soils.

The European IHME1500 map used to define the geological classes in the hydrological model (JAMS-J2000 + RF), with a scale of 1:1500000, shows 3 classes of geology in the Albarine catchment (karst, fine sediments, and coarse sediments). On the other hand, the French BD Charm-50 map (BRGM, 2020), with a scale of 1:50000, shows 71 different geological classes.

The RF is trained with the geological classes from the BD Charm-50 map to evaluate the impact of the precision of geological data in a catchment where flow intermittence is very influenced by the geological context. In this test, the JAMS-J2000 is still parameterized based on the IHME1500 map, only the input geology classes of the RF are modified.

## 3 Results

### 3.1 Observed flow state data analysis

Table 5 shows general statistics on the distribution of the observed flow state between the different datasets and the coverage of the river networks. The gauging stations are the main source of observed data in term of number of observations. They give information on long time periods with regular time step, but the number of station in the DRN is limited (1 station in the Genal and Lepsämänjoki DRNs and 5 stations in the Albarine DRN), which means that stations cannot bring useful information about the spatial patterns of drying in the DRNs. The field campaigns (with expertise) is the second source of observed data. They cover a shorter time period than the stations (3 months in Lepsämänjoki and 3.5 years in the Albarine), but have a better spatial coverage of the river network than the stations. In the Genal DRN, observed data from Google Earth images show a very good spatial coverage with about 38% of the river network with at least one observation along the period of available data. It also cover a long time period (11.5 years) but with only a few observation per reaches (between 1 and 8 observations per reach). Crowdsourced data only represent a very small fraction of the whole data (0.6 to 2.8%) but have a good spatial coverage, with around 14% of the Albarine and Genal river networks covered.





| DRN | Data type | Period of available data | Number of observations | Number of reaches with observations | Length of river network [%] |
|---|---|---|---|---|---|
| Albarine | Stations | 2005-10-01 to 2021-12-01 | 10681 | 5 | 1.2 |
| | Crowdsourced | 2019-06-26 to 2022-04-23 | 299 | 56 | 14.7 |
| | Field campaign | 2018-11-07 to 2022-04-30 | 5184 | 9 | 2.4 |
| | Expertise | 2018-11-07 to 2022-04-30 | 12688 | 10 | 1.9 |
| | **All** | **2005-10-01 to 2022-04-30** | **28852** | **61** | **15.7** |
| Genal | Stations | 2005-10-01 to 2021-10-19 | 5845 | 1 | 0.3 |
| | Crowdsourced | 2021-05-20 to 2022-02-12 | 88 | 28 | 14 |
| | Google Earth | 2010-10-08 to 2022-04-01 | 319 | 98 | 38.2 |
| | Expertise | 2021-03-28 to 2022-01-21 | 894 | 3 | 1.0 |
| | **All** | **2005-10-01 to 2022-04-01** | **7146** | **119** | **47.7** |
| Lepsämänjoki | Stations | 2005-10-01 to 2020-05-26 | 4761 | 1 | 0.1 |
| | Crowdsourced | 2021-06-18 to 2021-11-02 | 28 | 19 | 6.2 |
| | Field campaign | 2021-06-17 to 2021-09-26 | 807 | 8 | 2.6 |
| | Expertise | 2021-06-17 to 2021-09-26 | 711 | 7 | 1.6 |
| | **All** | **2005-10-01 to 2021-11-02** | **6307** | **23** | **7.3** |

**Table 5.** Flow observations in the studied DRNs (from 2005-10-01 to 2022-04-30)

Observed data have different distribution in time and space in the DRNs (Fig. 5a). For the three DRNs, there are observed data on the different classes of reaches (classified according to their drainage area), but there is more available data in the class of reaches with the largest drainage area. This is due to the gauging stations data that are located along the main river and which represent the largest share of the data. The Albarine basin is the only one to have a full seasonal coverage on the different types of reaches. Reaches with small drainage areas in the Genal and Lepsämänjoki DRNs have observed data mainly between June and September, and have missing data during the other months of the year (especially December and January). This shows that the collection of observed data on flow intermittence tends to be focused on the dry season and that there is almost no information on the state of flow of small river sections during winter.

Figure 5b shows the seasonal distribution of the dry observations along the river network. There is a clear spatio-temporal distribution of the dry observations in Lepsämänjoki DRN, with most of the drying events occurring in June and July in reaches with the smallest drainage area. Drying events gradually decrease with the size of the reaches drainage area, and the main river is perennial. However, drying events seems to be over-represented during the summer season, because in the smallest reaches, 100% of the observations are dry, whereas it is know that in this catchments, not all small reaches dry up, and they do not dry for more than a few weeks. In the Genal DRN, the peak of the drying season seems to be between June and September, but drying events are also observed in early spring and autumn. Most of the dry events are observed in the small reaches but a few




dry events are also observed in the downstream part of the Genal river due to water abstraction of irrigation (around 4% of the observations in June and July). Drying events are observed later in the season - from August to October - in the Albarine DRN

and are localized in small reaches but also in the main river due to the seepage of the Albarine river in the soil (around 30% of dry observations in the Albarine between July and August). The smallest reaches (with a drainage area lower than the 25th percentile only show flowing observation, which shows that dry observations may be lacking in these reaches.

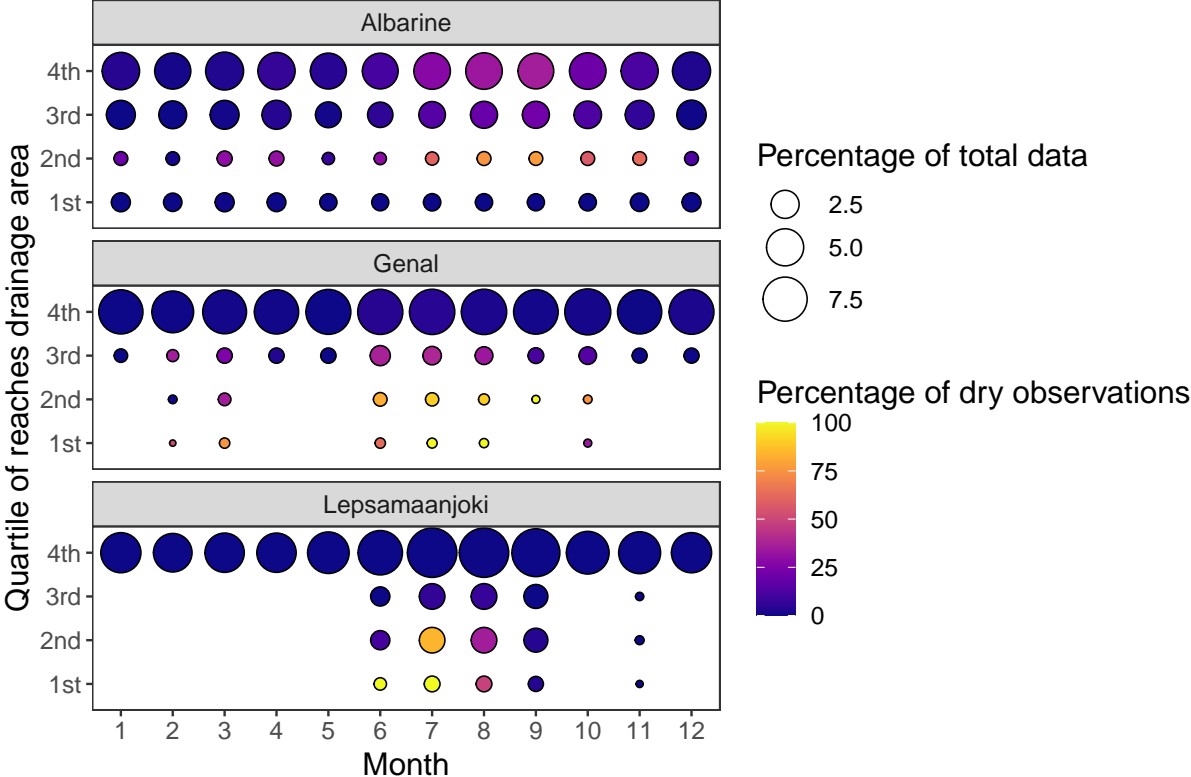

**Figure 5.** Distribution in space and time of flow state data. The size of the dots indicates the percentage of total available data per month and per class of drainage area, and the color the percentage of dry observations per month and class of drainage area.

### 3.2 Prediction of flow intermittence

This section presents the results of the simulation of flow intermittence with the JAMS-J2000+RF modelling.

Figure 6 shows an example of the state of flow prediction in one reach of the Albarine DRN. The comparison between the observed state of flow and the discharges simulated with the JAMS-J2000 model show that the hydrological model alone is not sufficient reproduce the periods with no flow. The transition from a flowing to a dry state cannot be easily inferred from the simulated flows alone since there are periods when the simulated discharge is relatively high while the phototrap indicates a dry state whereas on other periods, the simulated discharge is low while the phototrap indicates a flowing state. However, the



flow state predicted by the RF model is in good agreement with the observed flow states, which shows the which shows the usefulness of the coupling between the spatialized hydrological JAMS-J2000 model and the RF model.

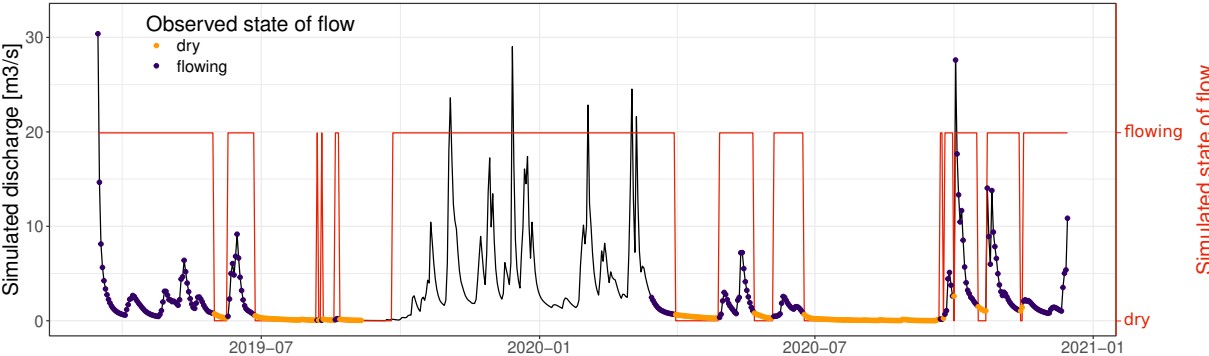

**Figure 6.** Daily state of flow predicted by the RF model (red) in the reach 2443600 in the Albarine DRN compared to the discharge simulated by JAMS-J2000 model (black) and the observed state of flow collected from a phototrap (orange: dry, purple: flowing).

In order to estimate more accurately the ability of the coupled JAMS-J2000+RF model to represent intermittency in whole river systems, the model was trained and tested with two different configurations: RF model is trained with 100% of the observed data (configuration 0) and trained again with 75% of the observed data and then validated on the remaining 25%
(configuration 1). The POD and FAR value obtained with the reduced training sample are an indicator of the RF model error to extrapolate the prediction of the state of flow on reaches and dates that are nor represented in the training dataset. With the configuration 0, the model perfectly reproduces the observed drying and flowing events in the three DRNs (POD = 100% and FAR = 0%), whereas the performance of the RF model is decreased with the configuration 1 (Fig. 7). The Albarine and Lepsämänjoki DRNs only show a slight decrease of the performance, the model still correctly predicts more than 90% of the
dry observations and has a FAR around 5%. The Genal DRN is more impacted by the removal of some of the observed data, the median POD drops to 65% and the median FAR is 19%. The results show that there is a high confidence of the prediction of the general dynamics of drying in the Albarine and Lepsämänjoki DRNs, and that the uncertainty is higher for the Genal DRN.

The results presented in the next sections of this study were obtained with the configuration 0.




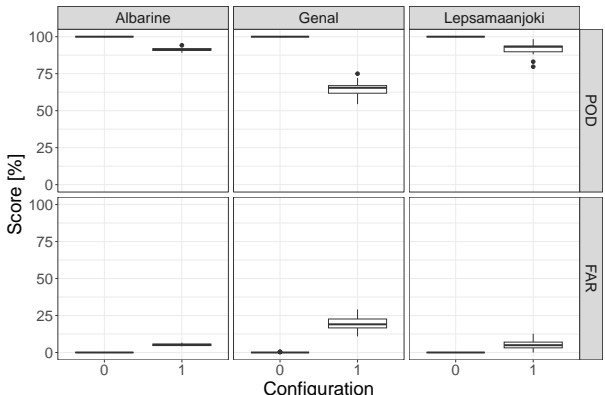

**Figure 7.** Performance of the RF model to predict dry events when the RF model is trained with 100% (configuration 0) and with 75% (configuration 1) of the observed data. Upper panel POD: probability of detection, lower panel FAR: False alarm ratio.

### 3.3 Simulated spatial and seasonal patterns of flow intermittence

Regarding the spatial pattern of flow intermittence, the model simulates more drying in the small tributaries for the three DRNs (Fig. 8). For the Albarine and Genal DRNs, the flow intermittence of the main river in the downstream part of the catchment, due to seepage for the Albarine and water abstraction for irrigation in Genal, is well reproduced by the model. Simulated spatial patterns of drying have been validated by local experts who confirmed that they are consistent with their observations (Fig. S13 and S14).

Figure 9 shows the mean interannual variations of the fraction of dry river network through the year. It shows that the drying is limited to end of May until end of August in the Lepsämänjoki DRN, and than the mean annual maximum of drying usually does not exceed 9% of the river network. In the Albarine DRN, the mean annual maximum of drying occurs in early September with between 24 and 27% of dry river network. More than 10% of the river network is continuously dry between July and the end of September. The model predicts some flow intermittence throughout the year (between 1 and 4% of dry river network during the winter season). In the Genal DRN, the river network can dry up to 78-80% in August, and more than 50% of the river network is dry from June to mid-September. The fraction of dry river network in Genal during the winter season stays relatively high (between 6 and 26%), but the lack of observed data over this period makes the results particularly uncertain.

Overall, the model successfully represents the general spatio-temporal patterns of drying in the 3 contrasted European DRN, with intense and long periods of drying in the Genal catchment characterized by a dry and warm climate, regular and localized drying up due to the geological context in the Albarine catchment, and short and limited in space drying in the Lepsämänjoki catchment characterized by a more humid climate but mild in summer.





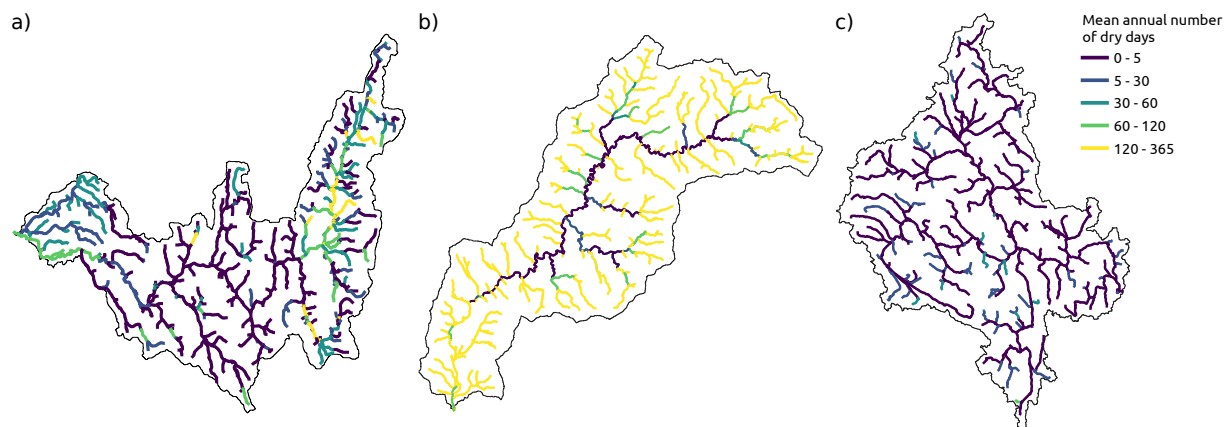

**Figure 8.** Predicted average annual number of dry days for each reach of the a) Albarine, b) Genal, c) Lepsämänjoki DRNs.

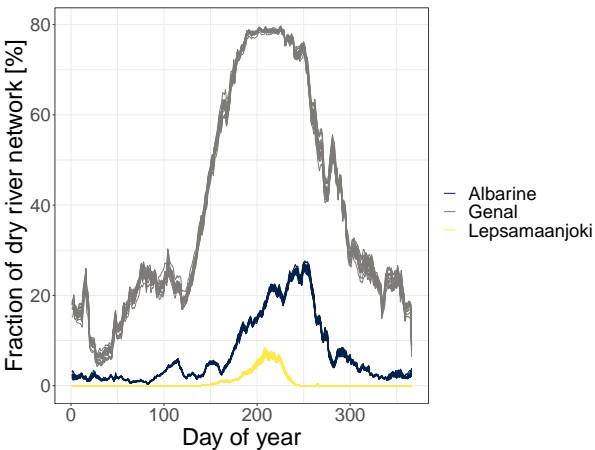

**Figure 9.** Seasonal variability of the fraction river network that gets dry (inter-annual average of the percentage of total number of kilometers of rivers). For each DRN, the lines represent the ensemble of the 20 runs of the RF model.

## 3.4 Analysis of the covariates

The ranking of the most important covariates in the RF models reflects the different contexts of flow intermittence in the DRNs. In the DRNs with more complexe spatial patterns of drying, the RF gives more weight to the variables describing the reaches characteristics. For all three DRNs, the drainage area of the reaches and their slopes are the 2 most important variables for the prediction of the flow state (Fig. 10).

For the Lepsämänjoki DRN, the next most important variables are the mean catchment air temperature during the previous 30 days (T30), the simulated discharge, and simulated groundwater contribution to the discharge (GW10 and GW). These three variables give information on the hydro-meteorological situation in the catchment, and define the temporal variability of




drying. T30 allows seasonal variability to be captured, and makes a distinction between winter low flows, when precipitation is stored as snow in the basin, and summer low flows, when drying is observed in small streams.

For the Genal DRN, the third and fourth most important covariates are the mean discharge during the 10 previous days and the current discharge, which shows that the temporal dynamics of drying is mainly controlled by the simulated discharge in the

reaches. The fifth most important variable is the landuse which reflects the more concentrated agricultural areas, with a water demand for irrigation, in the downstream part of the basin.

In the Albarine DRN, the most important variables, after the reaches drainage area and slope, are the landuse and soil types around the reaches and the current discharge. The four most important variables do not reflect the main cause of drying in the Albarine, which is the seepage of the river in moraine deposit areas. The classes of geology causing flow intermittence in the

Albarine are not represented in the IHME1500 dataset, which may explains why other spatial characteristics are used in the RF model to reproduce the spatial pattern of drying.

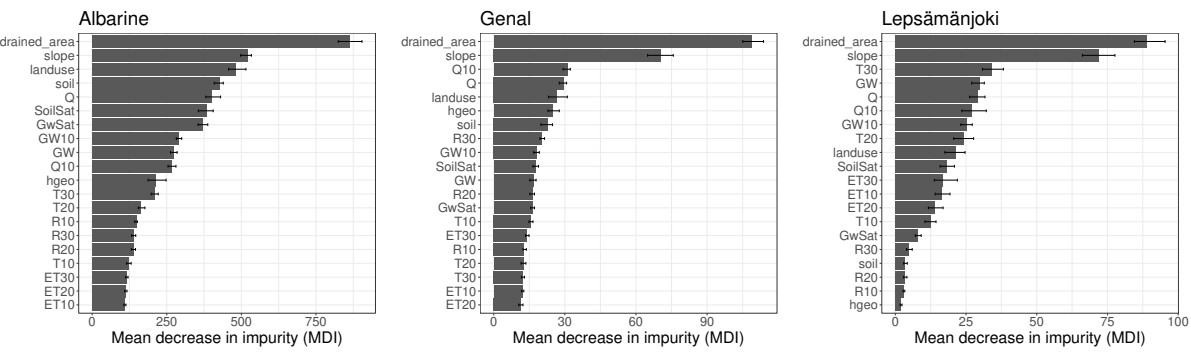

**Figure 10.** Importance of the covariates in the RF models (mean decrease in impurity (Archer and Kimes, 2008)) for the 3 DRNs. Bars represent the mean MDI and the error bars the minimum and maximum values of MDI for the 20 runs of the RF model.

## 3.5 Sensitivity to the input data

### 3.5.1 Sensitivity to the size of the training sample

Figure 11 shows the impact of the size of the training sample on simulated seasonal pattern of drying in the DRNs. The

RF model is either trained with 100% of the available observed data (configuration 0), or 75% of the observed data (configuration 1). For the Lepsämänjoki DRN there is no visible impact of reducing the training dataset on the predicted flow intermittence. In the Albarine and Genal DRNs, the results show that the uncertainty increases particularly during the winter season, when there are fewer observations.

These results show that the RF model is more sensitive to the representativity of drying in the observed data recorded than

in the amount of data itself. The Lepsämänjoki DRN has fewer observations and a poorer spatial and temporal coverage of the observed data than the Genal DRN, but the model is more robust in Lepsämänjoki than in Genal. The higher sensitivity of the Genal DRN to the training dataset can be explained by the fact that the DRN is more affected by drying, a very large part of





the river network dries every year, and during long periods (several weeks to several months). It thus needs a larger amount of observed data to fully capture the seasonal dynamics of drying along the river network.

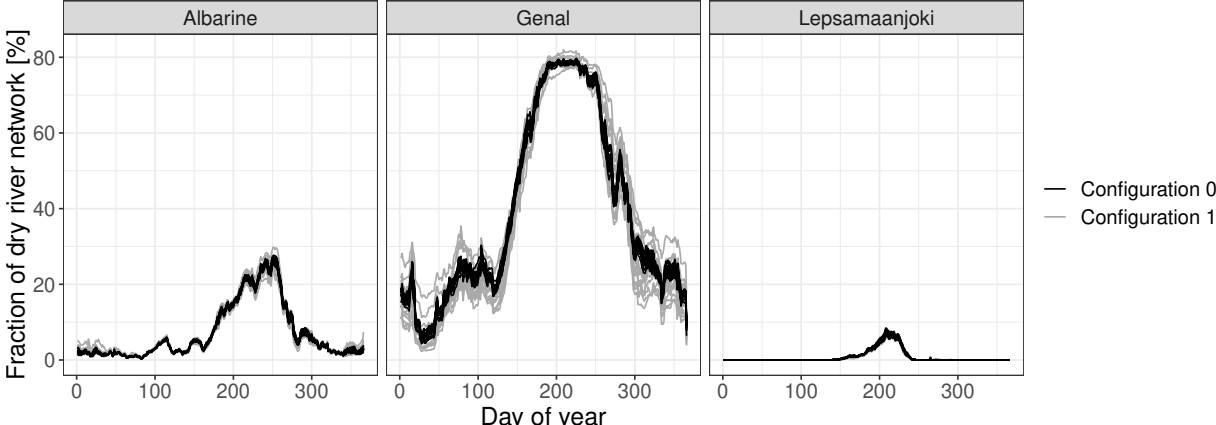

**Figure 11.** Sensitivity of the simulated length of dry river network to the size of the raining sample. Configuration 0 (black): the RF model is trained with 100% of the observed data. Configuration 1 (grey): the RF model is trained with 75% of the observed data.

### 330    3.5.2    Sensitivity to the type of flow state observed data

Figures 12 and 13 show the sensitivity of the model to the type of observed data used to train the RF. The first result is that the prediction of flow intermittence is very sensitive to the expertise data. Indeed, when this dataset is removed from the training sample, the FAR increases (43% for Albarine, 40% for Genal, and 7% for Lepsämänjoki), the drying is more intense during the summer in Lepsämänjoki (maximum annual of dry fraction of the river network between 6 and 8% without expertise data

versus 8 to 11% with expertise), and the drying is twice more intense and last much longer in the Albarine DRN.

Field campaign data also have a large impact on the prediction of flow intermittence, especially in the Albarine DRN where the model is only able to predict 50% of the dry days without the field data. The drying is much reduced during the summer and there is no drying simulated from November to June. On the opposite, in the Lepsämänjoki DRN, the FAR is increased without the field campaign data and the drying is very over-estimated. Expertise and field campaign data are the two most impactful

datasets in Albarine and Lepsämänjoki.

In the Genal DRN, the results show that the simulated seasonal pattern of drying is very different without the Google Earth data, with a lot of drying predicted during the winter season, which can reach unrealistic values (up to 70% of dry river network in January) (Fig. 13). In a DRN characterised by high intermittence of flows, and with little field observations, flow intermittence observation from remote sensing dataset can be very useful to better constrain the RF model.

In the Lepsämänjoki DRN, the removal of the stations data from the training dataset does not impact the prediction of drying. However, in the Albarine and Genal DRNs some of the stations are located on intermittent reaches and their removal decreases the POD of drying events to 61% for the Albarine, and 77% for Genal.



Crowdsourced data, which represents at most 1% of all observations collected in the DRNs, have a visible impact on the prediction of dryness, especially during the summer. For the Albarine, the mean annual maximum of dry river network decreases
from 27 to 23% without the crowdsourced data in early September. In the Genal DRN, the uncertainty increases without the crowdsourced data, for example, in late July - early August, the fraction of dry river network ranges between 78 and 80% when the RF model is trained with all of the observed data, and ranges between 78 and 85% when the crowdsourced data is removed from the training dataset. This shows that, even is they only represent a very small fraction of the observed data, crowdsourced data, have a significant impact on the prediction of flow intermittence through the spatial information they provide.

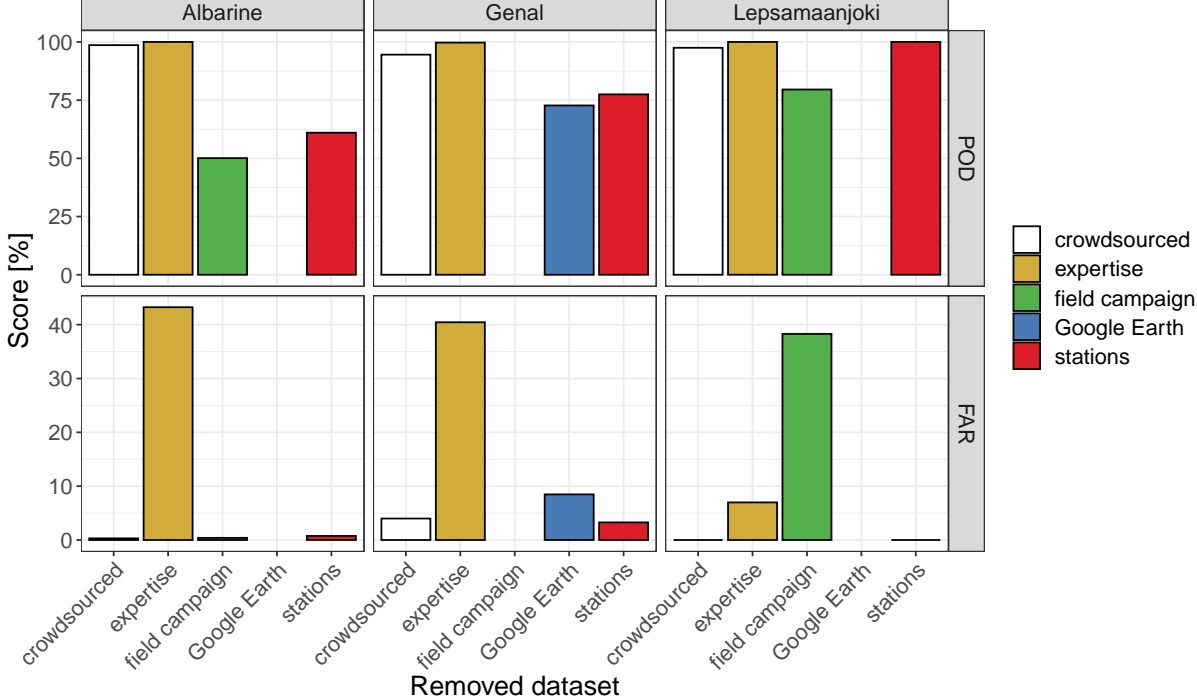

**Figure 12.** Impact of removing a source of observed data from the training sample on the performance of the RF model (upper panel POD: probability of detection, lower panel FAR: False alarm ratio).



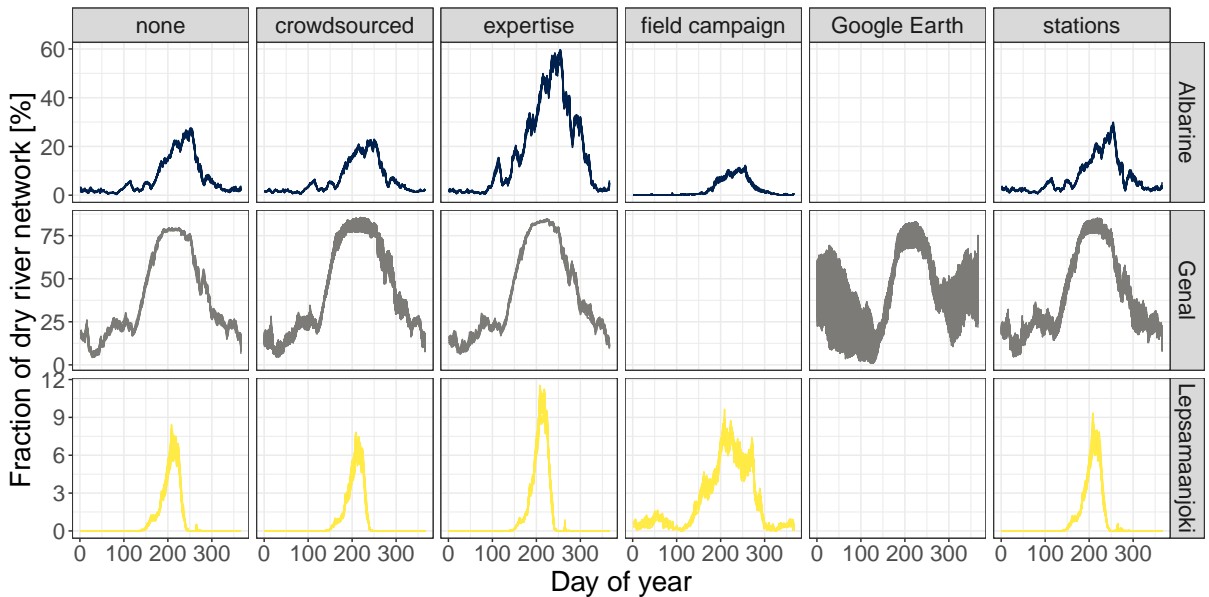

**Figure 13.** Impact of removing a source of observed data from the training sample on the prediction of flow intermittence.

### 3.5.3   Sensitivity to the geology data

When the BD CHARM-50 geology map is used to define the geology classes in the covariates of the RF model, geology becomes the most important variable in the RF (versus 11th most important variable with IHME1500) (Fig. 15). The RF model also gives more weight to the mean catchment ground water and soil saturation, which shows that the physical processes causing flow intermittence in the Albarine DRN are better taken into account in the RF model when using more accurate geological data.

The seasonal pattern of drying is rather similar but with a bit less drying in winter and spring and a bit more drying in summer and autumn. When looking at the spatial patterns of drying we can see some differences, especially in the upstream part of the catchment where there is moraine deposit that is not listed in IHME1500. With the coarser geology map (IHME1500), the RF manages to predict rather accurately the main spatio-temporal patterns of drying, but the use of more detailed geology map (BD CHARM-50) can help improving the prediction of drying at the reach scale.



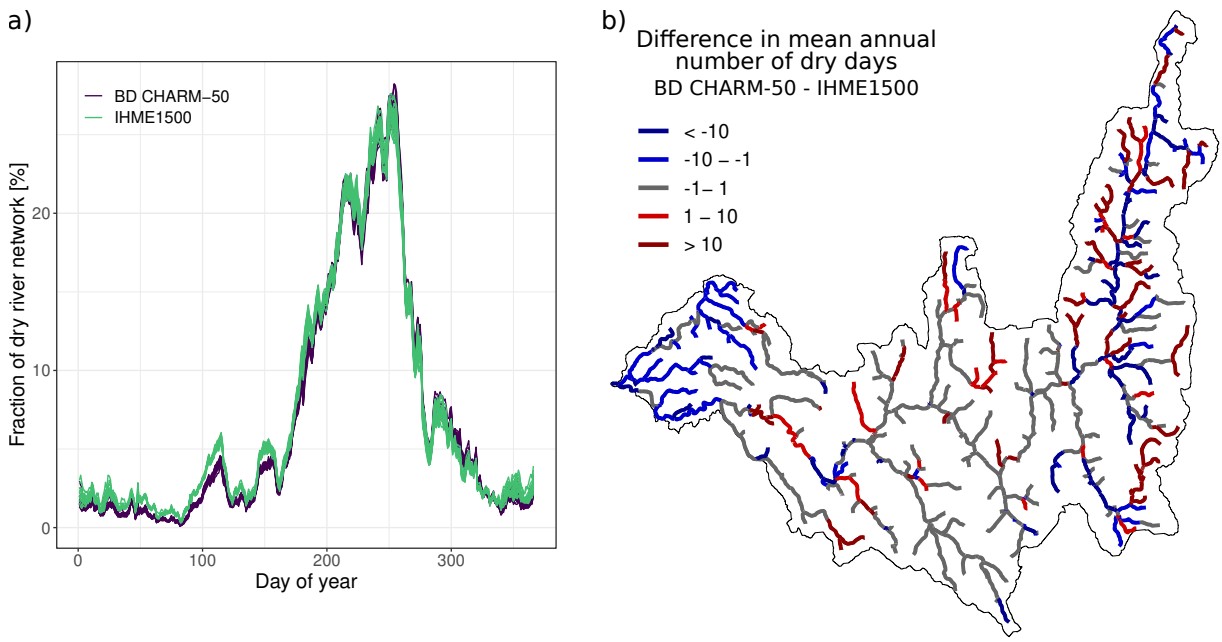

**Figure 14.** Sensitivity of the prediction of flow intermittence to geological data in the Albarine DRN.

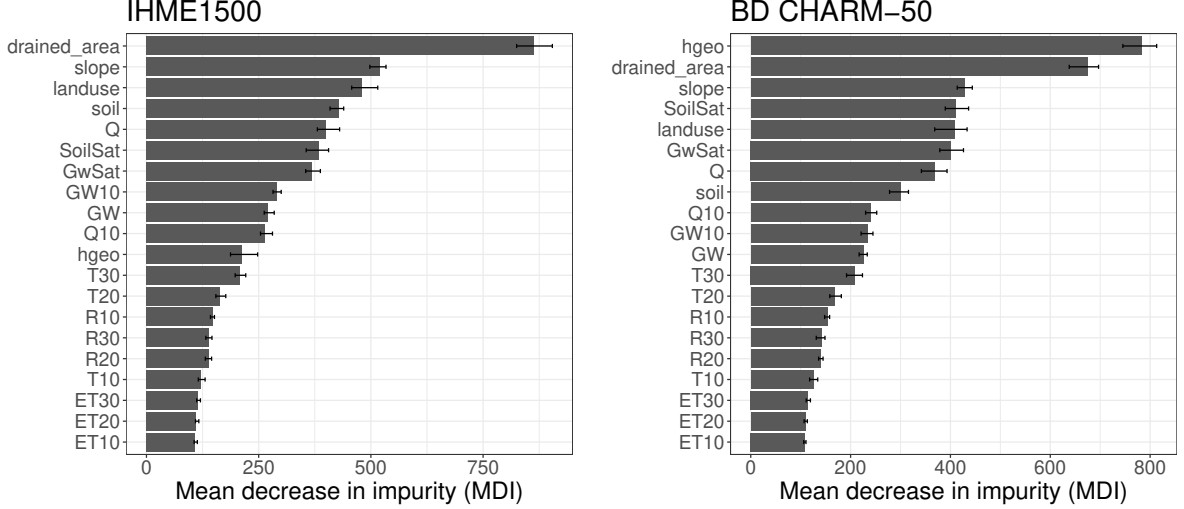

**Figure 15.** Importance of the covariates (mean decrease in impurity (Archer and Kimes, 2008)) when the RF model is trained with the IHME (left) and the BD CHARM-50 (right) geology maps. Bars represent the mean MDI and the error bars the minimum and maximum values of MDI for the 20 runs of the RF model.





## 4 Discussion

### 4.1 Hybrid modelling to predict flow intermittence at the reach scale

The coupling between a spatially distributed model and a random forest model has a number of benefits for predicting intermittency in river systems. First, the JAMS-J2000 model represents the spatially distributed hydrological physical processes in the catchments. This enables to simulate several hydrological variables at the HRU and the reach scale, such as evapotranspiration, soil water content, groundwater level, discharges, that can be used as spatially distributed covariates in the RF model. Second, the JAMS-J2000 represents lateral flow routing between the HRUs and the reaches, and thus represents the hydrological connectivity which cannot be represented in a RF model. However, the simulation of flow intermittence with JAMS-J2000 alone is not yet possible. The JAMS-J2000 model has difficulties to simulate periods with no flow. Even after long periods without precipitation input, the model tends to simulate residual low flows, and the reaches never completely dry up. There are also a multitude of processes causing the drying of rivers (e.g. interaction between the riverbed and the water table, seepage into karst, pumping of water from aquifers and rivers) and it is difficult to represent them all and accurately in a physical model (Fovet et al., 2021; Shanafield et al., 2021). Despite the JAMS-J2000 model ability of simulating seepage through the alluvial river bed (Watson et al., 2021), or water abstraction for anthropogenic uses (Branger et al., 2016), the data needed to parameterize these processes are seldom available and were not available in our case studies (e.g. daily amounts of water withdrawals and their precise locations). The use of the RF model enables to simulate flow intermittence even if the processes causing the drying up are are known or understood precisely beforehand since it does not require a representation of physical processes, but links covariates to observed states of flow. In addition, RF models have the advantage to provide variable importance metrics (Tyralis et al., 2019) which, in our case, allow to better understand the processes leading to the drying in the DRNs. The coupling of the two types of models is therefore very advantageous in order to be able to simulate flow states at fine time steps and in a spatialized way over the entire river network.

However, the use of a RF model has several limitations. A first limitation is that the RF model can predict the right state of flow for the wrong reasons if the causes of drying are not represented in the covariates. For the 3 studied DRNs, drainage area is the most important covariate, which is consistent with other studies using RF models to predict flow intermittence (Jaeger et al., 2023; González-Ferreras and Barquín, 2017; Snelder et al., 2013), but in the Albarine and Genal DRNs we know that the drying is in fact due to the geology and water abstraction, respectively. The results of the RF model does not necessarily provide a better understanding of the origin of drying in river networks if the covariates are not sufficiently precise. Most importantly, this means that a RF model trained on a specific DRN may not be robust enough to predict flow intermittence in another DRN.

One major application of this flow intermittence modelling approach is to simulate the flow states under different climate change scenarios and predict tipping points in the flow regime of the river sections, such as transitions from a perennial to an intermittent flow regime. However, the robustness of such a model for extrapolating flow intermittence in climate change projections is questionable. The RF model is trained with observed data over a relatively short period, with no observed change in the flow regime of the reaches and it is known that RF models cannot predict events that has never been observed before (Hengl et al., 2018; Tyralis et al., 2019), which represents a major limitation for predicting the future evolution of drying spells



in the DRNs. While it can be expected that the drying spells of currently intermittent reaches will be prolonged under climate change scenarios, the ability of the RF model to predict a shift from a perennial to an intermittent flow regime is not assured. However, the results of this study show that the average annual number of dry days simulated for the reaches known to have perennial flow is rarely zero, but can vary between 0 and 3 days per year. This means that in the present period, the model simulates completely perennial flow only in a few reaches. This bias in predicting the state of flow in the present period is a

drawback for characterising current drying dynamics in river systems and studying the impact on biodiversity, but may facilitate the prediction of drying in the context of climate projections as most of the reaches are already considered as intermittent in the present period.

### 4.2 Observed flow state data for the modelling of flow intermittence

The results of the RF model are highly dependent of the training dataset. This study highlights the challenges of obtaining

observed flow state data to train or validate the models. To accurately represent flow intermittence along river networks, the observed data ideally need to be uniformly distributed both spatially and temporally, which can be difficult to achieve.

Most studies focusing on the catchment scale collect observations from field campaigns (e.g., Llanos-Paez et al., 2023; Jaeger et al., 2023; Van Meerveld et al., 2019; Sefton et al., 2019), but such surveys generally do not allow rivers to be monitored over many years and are usually limited to portions of the river network as they can be very time-consuming.

This study shows the interest of combining different types of data with heterogeneous spatial and temporal patterns in order to maximise the information on flow condition in the river networks. This is consistent with the results of Gallart et al. (2016) who showed that combining data from citizen science and aerial photographs afforded more robust information.

The results obtained in the 3 basins demonstrate the need to adapt the data collection to the context of each DRN. Ideally, a large amount of homogeneously distributed data along the river network and throughout the year will introduce the least

possible bias into the model, like in the Albarine DRN for instance. However, the case of the Lepsämänjoki DRN shows that even with a small amount of data concentrated on the summer season, the predicted patterns of drying are consistent with the observations made by local experts. In contrast, with a similar amount of data, the variability of the prediction of flow intermittence in the Genal DRN is higher due to more complex spatio-temporal patterns of drying. To reduce the uncertainty in the Genal DRN, more years of observed data would be necessary, with data more evenly spread over the year to better capture

the length of dry spells.

The analysis of the Albarine and Lepsämänjoki DRNs shows that data from field campaigns provide essential information on the spatial and temporal dynamics of drying, making it the most useful type of data for predicting intermittency in river networks. However, in the Genal DRN, where phototraps were not installed during field campaigns, remote sensing seems to be a good alternative for collecting data. Although remote sensing data can be used to detect the state of flow adequately, Gallart

et al. (2016) have nevertheless pointed out several limitations: images are available at too low a frequency to study temporal patterns and dense vegetation near the rivers may prevent the detection of the state of flow.

As shown in the results of this study, citizen science can also be a useful way of obtaining intermittence data and increasing the spatial coverage of observations. Several studies have shown the advantages of working with citizens to monitor temporary



streams, especially to obtain observations in streams that would otherwise not be monitored (Turner and Richter, 2011; Buytaert
et al., 2014; Gallart et al., 2016; Kampf et al., 2018). Gallart et al. (2016) and Strobl et al. (2019) studied the accuracy of data
provided by citizen scientists and showed these data give an overall good indication of the hydrological state of the streams.

Expertise data indicating reaches with perennial flow proved to be crucial in reducing the over-representation of data from
intermittent reaches in the RF model training data across all three DRNs. However, this raises questions about the value of such
data, which is based on human perception and the error it may contain.

The general indications for data collection emerging from this study are to 1) favour a good spatial distribution of the obser-
vations by collecting data reaches with different characteristics (e.g. in terms of drainage area, geology and water abstractions),
2) collect data on intermittent sections as well as on reaches with a permanent flow regime, 3) have time series of observations
covering at least a whole year on a few points of the river network.

### 4.3 Delineation of the river networks

Another limitation of the study arises from the delineation of river networks. The delineation needs to be as accurate as possible
to ensure that observations of flow state are assigned to the correct reaches. However, several studies such as Prancevic and
Kirchner (2019); Van Meerveld et al. (2019); Godsey and Kirchner (2014) have shown that river networks are dynamic systems:
they extend or retract according to landscapes and climatic conditions and can also be disconnected. It is therefore difficult to
delineate a fixed reference river network with a density enabling to predict accurately the spatial variability of drying in the
DRNs.

In this study, the density of the delineated river networks was chosen so that all observations could be assigned to a reach,
but the results show that the density of the river network have an impact on the simulated patterns of drying in the DRNs. In
the 3 studied DRNs, contradictory states of flow were observed in reaches at the same day indicating that the density of the
river networks is not high enough to capture very local processes of drying. In contrast, the density of the river network should
not be too high, as it may lead to the representation of reaches with an unrealistically small drained area, for which there is no
observed data available to train the RF model. This situation occurred in the Albarine DRN where the resolution of the river
network had to be increased in order to capture the locations of observed data (Fig. S2), resulting in some unrealistic prediction
of small perennial reaches in the upstream part of the catchment (Fig. 8).

### 5 Conclusions and perspectives

The modelling approach, coupling a spatially distributed physical hydrological model (JAMS-J2000) with a random forest
classification model, developed in this study allows to predict the daily state of flow (dry or flowing) at the reach scale along
river networks. The results show that the models allow to successfully predict the main spatio-temporal patterns of drying in 3
contrasted European river networks.

This study also discusses the difficulty of collecting flow intermittency data to train and validate random forest models. The
results show that the combination of various sources of observed flow state data is essential to form a training dataset that is



representative of the actual spatio-temporal drying patterns in the drying river networks and to reduce the uncertainty of the prediction of flow intermittence.

In order to improve the modelling of flow intermittence, further improvements could be made to the models and to the collection of flow state data to train the RF model. Regarding the modelling approach, a first perspective is to add a third class
of state of flow in the RF model to predict the pools condition (i.e. stagnant water in disconnected pools) which is as important as the dry or flowing conditions for studying the ecological impact of flow intermittence Datry et al. (2017); Bourke et al. (2023). Another perspective is to improve the parameterization of the groundwater reservoir in the JAMS-J2000 models by using observed data of groundwater level to optimize the groundwater parameters and by using a more precise geology data to define the geological classes in JAMS-J2000 for the DRNs where flow intermittence is influenced by geology. Regarding the
collection of flow state data, a perspective is to use satellite products to collect flow intermittence data. Cavallo et al. (2022) showed that Sentinel-2 images can be used to detect flow intermittence along river networks. The use of satellite products could allow the modelling method to be transposed more easily to other river networks without the need for extensive field campaigns.

The hydrological modelling approach presented in this study will be used to project the evolution of flow intermittence in
the river networks under climate change scenarios and provide flow intermittence indices to characterize the spatio-temporal dynamics of drying in the DRNs in the present and future periods. These indices will then be used to study the impact of drying on the freshwater ecosystems.

*Code and data availability.* The calibrated JAMS-J2000 hydrological models for the 3 study catchments, the R scripts used to predict flow intermittency with a Random Forest algorithm, as well as the observed flow state data used in this study an be obtained from the corresponding
author upon request.

*Author contributions.* Conceptualization: LM, AK; model implementation and analysis: LM, AK, AD; draft preparation and discussions: LM, FB, JPV, AK; All authors read and approved the final paper.

*Competing interests.* The contact author has declared that none of the authors has any competing interests

*Acknowledgements.* This study was supported by the H2020 European Research and Innovation action Grant Agreement no. 869226
(DRYvER). We thank Thibault Datry and Bertrand Launay (INRAE RiverLy) for their expertise on the Albarine DRN, Heikki Mykrä and Henna Snåre (SYKE) for their expertise on the Lepsämänjoki DRN, and Nuria Bonada, Maria Soria (University of Barcelona), Amaia Angula Rodeles (Universidad de Cantabria) and Nuria Cid (INRAE) for their expertise on the Genal DRN. We also thank all the other members of DRN Teams of the DRYvER project for sharing local data and collecting flow intermittence observations in the DRNs.



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
