# Peer review of "Flow intermittence prediction using a hybrid hydrological modelling approach: influence of observed intermittence data on the training of a random forest model"

_EGUsphere, 2023_

## Author Comment (AC2)

**Line 128 In cases of simultaneous acquisitions from multiple resources. Was the information coincident? For example, did information from traditional measurement stations coincide with google images? Was the 'flow condition information the same?**

By grouping the data by reach and by date, we observe that there is simultaneity in only 0.26% of cases on average for the 3 catchments (Albarine: 83 cases of simulatneity over 28852 total cases, Genal: 16 cases over 7146, Lepsamanjoki: 12 cases over 6307).
The small amount of data observed on the same day on the same stretch of river can be explained by the complementary nature of the different sources, which each focus on different areas and periods.
Of the 111 cases of simultaneity, the different sources give the same state of flow in 88% of cases. Figure A shows the number of cases for which the sources agree or disagree according to the number of sources.

[Figure]

*Figure A: Number of cases (reach and dates) with simultaneous observations*

Disagreements between data sources may be due to :
- local phenomena: it is possible to observe different states of flow along the same reach;
- the subjectivity of observers during data acquisition (particularly in cases where there is stagnant water or very low flow) ;
- errors in assigning the flow class.

**Line 200  I suggest you use accuracy, sensitivity, specifcity, and true skill statistic to validated RF**

As you suggested we computed accuracy, sensitivity, specifcity, and true skill statistic (TSS) to validate RF. Sensitivity is équivalent to POD (probability of prediction of a drying event) which was originaly used in the manuscript. Along with Sensitivity, Specificity, Accuracy and TSS, we kept the False Alarm ratio (FAR) which indicates the probability of predicting a false drying event. Figure B shows that Specificity is above 0.99 for the 3 catchments, which means that the RF model predicts flowing events almost perfectly. Accuracy is also very close to 1 (> 0.98), this is due to the fact that flowing events are much more represented than drying events in the observed dataset, so prediction errors for dry events are negligible compared with the near-perfect predictions of flowing events. TSS (which is computed as Sensitivity + Specificity -1) is very similar to Sensitivity as Specificity is very close to 1.

[Figure]

*Figure B :Performance of the RF model when the model is trained with 75% of observed data and tested on the remaining 25% (configuration 1). TSS: True Skill Statistic, FAR: False Alarm Ratio (this figure will replace Fig.7 in the manuscript)*

These new elements will be integrated to the revised version of the manuscript.

---

## Author Comment (AC3)

**6. It would be relevant to include the annual mean temperature and total precipitation in table 1 or in the study areas description section. These would help better understand the climate contrast among the different study sites.**

Thank you for this suggestion, here are the mean annual temperature and precipitation for the 3 catchments (from ERA5-land reanalysis during the period 1991-2020) :

| Catchment | Mean annual temperature (°C) | Mean annual precipitation (mm) |
|---|---|---|
| Albarine (FR) | 10.0 | 1439 |
| Genal (ES) | 15.9 | 743 |
| Lepsämänjoki (FI) | 5.6 | 899 |

**12. Line 157. Not clear if the period that you mention is for all study sites or whether it is for the calibration or validation step. Please clarify.**

After the steps of calibration and validation, the JAMS/J2000 is used to reconstruct daily hydrological variables (discharge, groundwater contributions, evapotranspiration, snowmelt, soil saturation and groundwater saturation) for the period 2005-10-01 to 2022-04-30 for the 3 study sites.

**14. Table 2. The Albarine and Lepsämänjoki sites have a continuous simulation period but the Genal has a gap between the calibration and validation period. Would it be relevant to explain why is that?**

We encountered inconsistencies within the measured discharge data that raised concerns about the data accuracy. We analyzed the model performance in adjacent watersheds and found that the model performed well in the entire region. Hence, we concluded to leave out the periods, which are most certainly indicating potential errors. In an effort to maintain a collaborative and respectful relationship with the data providers, we aimed to handle the observed inconsistencies delicately and without causing undue concern. However, we acknowledge the validity of your point that mentioning these inconsistencies is crucial for transparency and accuracy in our research analysis. Therefore, we will mention this point in our final version.

**15. The metrics in table 3 could also include a metric similar to your POD and FAR but for the hydrological model. For instance, set a low-flow threshold and indicate how many times the model succeeds simulating when the river is below the threshold (POD) or when the model is not simulating it accurately (FAR). This would also give information on how good the hydrological model is to simulate low-flow periods, which I think would add relevant information for your study. Is the hydrological model skillful simulating low-flow periods or is the RF algorithm making most of the work?**

As you suggested, we analysed the performance of the JAMS/J2000 model to simulate discharges below a defined threshold at the gauging stations in the 3 catchments.
The chosen threshold is the 10th quantile of observed discharges on the total period (calibration and validation), keeping only the values for hydrological years with less than 10% missing values.

Then the performance metrics were computed as follows (the following metrics are different from the metrics used to analyse the performance of the RF model in the manuscript in response to a comment from referee RC2) :

Sensitivity = a / (a+c) (similar to Probability of Detection of drying events (POD) in the manuscript)
Specificity = b / (b+d) (similar to POD of flowing events)
Accuracy = (a+d) / (a+b+c+d) (probability of correctly simulating discharges below and over the 10th quantile)
True Skill Statistic (TSS) = Sensitivity + Specificity - 1

| Simulated discharges | Observed discharges | |
| --- | --- | --- |
| | Qobs < threshold | Qobs > threshold |
| Qsim < threshold | a | b |
| Qsim > threshold | c | d |

a, b, c, and d represent the number of days for each of the 4 conditions during the calibration or validation period of the JAMS/J2000 model.

The Figure 1 shows the results of the analysis of the performance of JAMS/J2000 to simulate low flows.
For the validation period, Sensitivity is respectively equal to 0.84, 0.62 and 0.91 for the St-Rambert (Albarine), Lepsämänjoki, and Jubrique (Genal) gauging stations, which shows that the hydrological model simulates low-flow periods very well (for Saint-Rambert and Jubrique) and fairly well (for Lepsämänjoki).
However, for the St-Denis gauging station in the Albarine catchment, Sensitivity is equal to 0. This is due to the fact that the river is intermittent and sometimes completely dries at this station. For the St-Denis station the 10th quantile is equal to 0 m3/s, and as the JAMS/J2000 is not able to simulate complete drying, the model performance is poor.
These results show that the JAMS/J2000 hydrological model can provide correct simulations of the alternation between periods of low flow and medium-high flow to the RF model, but that the J2000-RF coupling is essential for correctly simulating the flow on intermittent river sections.

[Figure]

*Figure A : Metrics for the prediction of low flows (< 10th quantile of observed discharges) with JAMS/J2000 model at the gauging stations for the calibration and validation periods.*

**17. Lines 167 and 278: Is it possible to infer why the uncertainty for the Genal catchment is higher? Is it that the catchment is very complex to simulate or some other issue?**

We cannot say with certainty why the uncertainty is greater for the Genal catchment in Spain, but our hypothesis is that ther is a lack of observed flow state data to properly characterise drying patterns in the catchment, which is more complex than in the 2 other catchments (possible drying up

throughout the year and widespread drying up throughout the catchment). There is approximately the same amount of observed data in the spanish and finnish catchments, but the flow intermittence pattern in the finnish catchment is less complex (drying only during the summer month and only on small tributaries). Besides, the elevation and hence, slope range is much higher in Genal, which leads to a more heterogeneous pattern of drying, which also demands more observed data to increase robustness.

**18. Line 279: Would it be relevant to also include some results for configuration 1 (perhaps in the supplement)? Just for comparison and to include something related to the uncertainty of the results.**

Section « 3.5.1 Sensitivity to the size of the training sample » and more specifically Figure 11 already shows a comparison of the seasonal patterns of drying with configurations 0 and 1.

We agree that the sentence line 279 « The results presented in the next sections of this study were obtained with the configuration 0. » is confusing and will be removed. We will make it clearer that the spatial and seasonal patterns of flow intermittence as well as the covariates are first analysed with configuration 0, and that the sensitivity to the size of the training sample (configuration 1) or the type of observed data is analysed next.

We did, however, try to include some more results with configuration 1 to give a better idea of the uncertainty related to the size of the training sample. Figure B shows the importance of the covariates with configuration 1 in the 3 catchments. It is very similar to Figure 10 from the manuscript (with configuration 0) for the 5 most important covariates, which shows that for this study the importance of the covariates is not very sensitive to the size of the training sample, but rather to the quality of the covariates (cf section 3.5.3 Sensitivity to the geology data).
Figure B can be included in the supplementary material of the revised manuscript.

*Fig*

[Figure]

*ure B :Importance of the covariates in the RF models (mean decrease in impurity (Archer and Kimes, 2008)) for the 3 DRNs. Bars represent the mean MDI and the error bars the minimum and maximum*

**20. Lines 356 to 260. It is not clear if this paragraph only refers to the Albarine catchment. Please clarify.**

Yes, lines 356 to 260 refer only to the Albarine catchment, as it is the only cachment for which we have another source of geological data. We will clarify this in the revised version of the manuscript.

**22. Line 439. You could support this claim using previous expert elicitation studies that looked into expert perception uncertainty. For example: https://doi.org/10.5194/hess-26-5605-2022 or https://doi.org/10.1002/2015WR018461**

Thank you for recommanding the 2 studies on expert elicitation. We propose adding the following sentences after line 439:

Expert elicitation in hydrology has already shown benefits, particularly when tangible data are missing (Ye et al, 2008, Warmink et al. 2011, Sebok et al. 2016, Sebok et al. 2022). These studies do show differences in the individual perceptions of the experts consulted, but by consulting a larger number of experts (in this study, only 1 or 2 experts were consulted per studied DRN) and by applying protocols similar to the ones proposed in these studies, the uncertainty linked to individual perception could be reduced, or at least quantified.

---

## Author Response (AR1)

Dear editor and referees,

We would like to thank you again for your positive comments and suggestions to improve the manuscript.
Please find below the point by point modifications brought to the revised manuscript in response to your suggestions.
Please note that the line numbers given below in our answers refer to the revised version of the manuscript.

With kind regards,
Louise Mimeau et al.

**Editor**

**- Provide a overview of what is meant by flow intermittence based on the various definitions in the literature, and provide a clear definition adopted for the current work.**

The following paragraph was added line 29:
« The term intermittent rivers refers to all rivers with a non-perennial flow. This includes ephemeral rivers with short periods of flow in direct response to rainfall or snow melt events, rivers with seasonal flow, and nearly perennial rivers with infrequent periods of drying (Buttle et al., 2012; Snelder et al., 2013; Shanafield et al., 2021).
In this study, the term "flow intermittence" will refer to the alternation between flowing phases and phases with interrupted flow (completely dried riverbed or disconnected pools). »

**- In section 2.2.3 is not exactly clear whether the RF is trained using modelled or observed data. Reading "the results of the JAMS-J2000 hydrological models are used to train a Machine Learning model" suggests that the RF is trained using modelled data. But then "For each DRN, the RF models are trained using flow state observations" suggests otherwise. Please clarify.**

We agree that this sentence is confusing, it was replaced with :
« Results of the JAMS-J2000 hydrological models are used as input data to a Machine Learning model to predict the flow intermittence at the reach level. » (Line 189)

**- In Section 3.2 it is unclear whether the model with configuration 1 is evaluated on the entire dataset, or on the 25% validation dataset. If the performance is evaluated on the entire dataset, the drop in performance appears significant. Please clarify.**

Lines 288 – 291 were replaced with «To enhance the precision of evaluating the coupled JAMS-J2000+RF model to represent intermittency across the entire river systems, the model underwent training and testing using two distinct configurations: Configuration 0 involved training the RF model with 100% of the observed data, while Configuration 1 involved training the RF model with 75% of the observed data and validating its performance on the remaining 25%. »

**- By looking at Figure 6, it would be interesting to see how the RF model would compare with a simple threshold applied to simulated discharge. E.g. if simulated discharge is less than a specified threshold, than assume no flow.**

The following paragraph (Lines 416 to 427) and Figure 16 were added to the discussion to compare the results obtained with the RF model to a simple threshold method.

« The question can be raised about the contribution of the RF model compared to applying a threshold on the discharges simulated JAMS-J2000 model below which zero flow would be determined. Figure16 shows the distributions of simulated discharges for the two types of flow conditions (flow or dry) in reach 2443600 of the Albarine (same example as in Figure 6).
For simulated discharges ranging from 0 to 4 m3/s, there is an intersection of the distributions for observed dry and flowing events. Setting a threshold would mean truncating the tails of these distributions. For instance, by setting a threshold to achieve a Sensitivity of 98% on this reach, a FAR of 26% is obtained, as low discharges are all predicted as dry events. In contrast, with the RF model, the intersection of the distributions is well reproduced, and a FAR of 1.7% is achieved for the same Sensitivity (98%). The differences in distributions between observed and simulated flow conditions can be explained by the fact that there are few "flowing" observations during winter periods with high flows in this reach.
It is also challenging to extrapolate a discharge threshold value across all reaches of the network. Looking at the spatial pattern of flow intermittence in the DRNs (Fig.8) it is clear that the threshold value should be spatially distributed to take account of local effects, but this raises the question of how this spatial distribution should be achieved. »

[Figure]

*Figure 16: Distribution of simulated discharge with JAMS-J2000 model for observed and simulated (RF model 20-members ensemble with configuration 1) flowing conditions for reach 2443600 in the Albarine DRN (same reach as in Fig.6). The horizontal black line shows an example of an applied threshold to predict flow condition from the simulated discharges (on the figure, the threshold value is set to correctly predict 98% of the observed drying events).*

**Referee RC2**

**Abstract: add a summary of model performance (quantitative values)**
The following sentence was added to the abstract (line 13):
« Results show that the hybrid modelling approach developed in this study allows to accurately predict the spatio-temporal patterns of drying in the 3 catchments, with a sensitivity criteria above 0.9 for the prediction of dry events in the Finnish and French case studies, and 0.65 for the Spanish case study. »

**Line 17 replace home with habitat**
Line 19 was replaced with : « They are habitat to many animal and plant species within the riverbed and in the riparian zone ».

**Line 20 "These ecological corridors can be disrupted when river beds dry up" I think this point should be expanded, it is true that some species cannot live in the presence of water, but non-perennial rivers due to the succession of different flow phases represent biodiversity hotspots.**
« These ecological corridors can be disrupted when river beds dry up. » was replaced by « In particular, ecologists assume that intermittent rivers are biodiversity hotspots thanks to the succession of different flow phases (e.g. flowing, isolated pools, dry) which promotes species richness (Datry et al., 2014) » (Line 22)

**Line 68 I think it is better recall this section "materials and methods"**
Section 2 was recalled « Material and methods » (Line 73).

**Line 128 In cases of simultaneous acquisitions from multiple resources. Was the information coincident? For example, did information from traditional measurement stations coincide with google images? Was the 'flow condition information the same?**
The following paragraph was inserted lines 132-137.
« As a result of acquiring data from multiple sources, there may be several flow state observations on the same day in the same reach. By grouping the data by reach and by date, we observe that there is simultaneity in only 0.26% of cases on average for the 3 catchments (Albarine: 83 cases of simultaneity over 28852 total cases, Genal: 16 cases over 7146, Lepsamanjoki: 12 cases over 6307). The small amount of data observed on the same day on the same stretch of river can be explained by the complementary nature of the different sources, which each focus on different areas and periods. Of the 111 cases of simultaneity, the different sources give the same state of flow in 88% of cases. »

**Line 200 I suggest you use accuracy, sensitivity, specificity, and true skill statistic to validated RF**
We replaced the POD and FAR criteria with sensitivity, specificity, accuracy and FAR.
We chose not to include True Skill Statistic (TSS) in the manuscript, as Specificity being very close to 1 for the 3 case studies, TSS is very similar to the sensitivity criteria and does not bring more information regarding the performance of the model. However, we chose to keep the FAR (false alarm ratio) as it shows information regarding the proportion of simulated dry events which may turn out to be false positives.

The changes made mainly concern lines 206 to 219 for the definition of the performance criteria, Figures 7 and 12 which were replaced to show the new criteria, as well as as their comments in sections 3.2 and 3.5.2.

**Referee RC3**

**1. Line 23: A word is missing in the sentence, perhaps "limit"? For example: … flow can endanger ecosystems and limit the access to water resources …".**
Line 26 was change to : « Prolonged drying and shifting of river sections from perennial to intermittent flow can endanger ecosystems and limit the access to water resources useful to our society (Steward et al., 2012; De Girolamo et al., 2017; Tonkin et al., 2019). »

**2. Line 46: I think that the "a" is extra in the beginning of the sentence: "Another a challenge in …".**
The extra « a » was removed » (Line 55).

**3. Line 51: Change "do" into "to".**
« do » was changed into « to » (Line 59).

**4. Lines 61 to 64: This paragraph is a repetition from the previous paragraph. Please remove one.**
This paragraph was removed (lines 61 to 64 of the first version of the manuscript).

**5. Line 74: Change the sentence to "…, located in the South of Spain..." or ",… located in southern Spain…".**
Line 79 was changed into : « The Genal catchment, located in southern Spain »

**6. It would be relevant to include the annual mean temperature and total precipitation in table 1 or in the study areas description section. These would help better understand the climate contrast among the different study sites.**
Annual temperature and precipitations were added to Table 1.

**7. Line 83: you could add "input to" and change "modelling" to "model" in the sentence. ".., hydrogeology information needed for input to the spatially distributed hydrological model…".**
Line 88 was changed into : « Topography, soil types, landuse and hydrogeology information is needed as input to the spatially distributed hydrological model. »

**8. Please use superscript for units throughout the text (e.g. cubic meters, square meters, etc.). Figure 6**
Units were corrected in Fig.6 as well as in the rest of the manuscript.

**9. Consider changing the symbology for the Google Earth observations because they are not easily observed as they can be confused with the river network.**
Symbology of Google Earth in Fig.2 was changed into black crosses to avoid confusions with the river network. Colors in Fig.12 were modified to fit the new symbology in Fig.2.

**10. In Figure 3. It should be "Daily flow state", a "t" is missing.**
The missing « t » in Fig.3 was corrected.

**11. Line 147. A period is missing after (HRU).**
The missing period was added Line 157.

**12. Line 157. Not clear if the period that you mention is for all study sites or whether it is for the calibration or validation step. Please clarify.**
In order to clarify, the sentence that was originally at the end of section 2.2.1 : « Daily hydro-meteorological variables such as spatially distributed discharges and groundwater contributions, as well as evapotranspiration, snowmelt, soil saturation and groundwater saturation at the catchment scale are simulated with the JAMS-J2000 model from 01/10/2005 to 30/04/2022. » was removed and the following sentence was added to the end of the « 2.2.2 Calibration of JAMS-J2000 model » section (Line 185) :
« Once calibrated, the JAMS-J2000 model is used to simulate daily hydro-meteorological variables such as spatially distributed discharges and groundwater contributions, as well as evapotranspiration, snowmelt, soil saturation and groundwater saturation at the catchment scale from 01/10/2005 to 30/04/2022 in the 3 studied DRNs »

**13. Figure 4: I recommend that you add the definitions of MPS, DPS, LPS and all the abbreviations that you used in the figure. These could go into the caption of the figure.**
The definitions of MPS, LPS, DPS were added to the caption of Fig.4.

**14. Table 2. The Albarine and Lepsämänjoki sites have a continuous simulation period but the Genal has a gap between the calibration and validation period. Would it be relevant to explain why is that?**
The following sentence was added Line 175 :

« For the Genal catchment, the discharge data measured at the Jubrique gauging station indicated potential errors between 2004 and 2012, this period was therefore not taken into account in the calibration and validation of the model. »

**15. The metrics in table 3 could also include a metric similar to your POD and FAR but for the hydrological model. For instance, set a low-flow threshold and indicate how many times the model succeeds simulating when the river is below the threshold (POD) or when the model is not simulating it accurately (FAR). This would also give information on how good the hydrological model is to simulate low-flow periods, which I think would add relevant information for your study. Is the hydrological model skillful simulating low-flow periods or is the RF algorithm making most of the work?**

To avoid making the manuscript too long, the following following sentence was added Line 183 : « More details on the validation of the JAMS-J2000 model on low flows are available in supplementary material. », and the following paragraph was added to the supplementary material.

« In this section we analyse the performance of the JAMS-J2000 model to simulate discharges below a defined threshold at the gauging stations in the 3 catchments. The threshold is taken as the 10th percentile of observed discharges on the total period (calibration and validation), keeping only the values for hydrological years with less than 10% missing values.

Then the performance metrics were computed as follows:

Sensitivity = a / (a+c)
Specificity = d / (b+d)
Accuracy = (a+d) / (a+b+c+d)
FAR = b / (a + b)

with a, b, c, and d the number of simulated days for each of the 4 conditions defined in the confusion matrix (Table S.5.).

|  | Observed discharges | |
| --- | --- | --- |
| Simulated discharges | Qobs < threshold | Qobs > threshold |
| Qsim < threshold | a | b |
| Qsim > threshold | c | d |

*Table S.5. : Confusion matrix of matches and mismatches of predictions and observations of low flows with JAMS-J2000.*

Figure S.13. shows the results of the analysis of the performance of JAMS-J2000 to simulate low flows.
For the validation period, Sensitivity is respectively equal to 0.84, 0.62 and 0.91 for the St-Rambert (Albarine), Lepsämänjoki, and Jubrique (Genal) gauging stations, which shows that the hydrological model simulates low-flow periods very well (for Saint-Rambert and Jubrique) and fairly well (for Lepsämänjoki).
However, for the St-Denis gauging station in the Albarine catchment, Sensitivity is equal to 0. This is due to the fact that the river is intermittent and sometimes completely dries at this station. For the St-Denis station the 10th percentile is equal to 0 m3/s, and as the JAMS-J2000 is not able to simulate complete drying, the model performance is poor.
These results show that the JAMS-J2000 hydrological model can provide correct simulations of the alternation between periods of low flow and medium-high flow to the RF model, but that the J2000-RF coupling is essential for correctly simulating the flow on intermittent river sections.

[Figure]

*Figure S.13. : Metrics for the prediction of low flows (< 10th percentile of observed discharges) with JAMS-J2000 model at the gauging stations for the calibration and validation periods. »*

**16. Figure 5. Along the text you refer to Fig. 5a or Fig. 5b, but in figure 5 there is no division between sections a or b.**
Lines 258 and 266, the reference to Figure 5 were corrected.

**17. Lines 167 and 278: Is it possible to infer why the uncertainty for the Genal catchment is higher? Is it that the catchment is very complex to simulate or some other issue?**
This point was already discussed in section 4.2 Line 463 of the manuscript :
« However, the case of the Lepsämänjoki DRN shows that even with a small amount of data concentrated on the summer season, the predicted patterns of drying are consistent with the observations made by local experts. In contrast, with a similar amount of data, the variability of the prediction of flow intermittence in the Genal DRN is higher due to more complex spatio-temporal patterns of drying. To reduce the uncertainty in the Genal DRN, more years of observed data would be necessary, with data more evenly spread over the year to better capture the length of dry spells. »

To make it easier to understand, we added Line 303 (section 3.2) « (see discussion in Section 4.) »

**18. Line 279: Would it be relevant to also include some results for configuration 1 (perhaps in the supplement)? Just for comparison and to include something related to the uncertainty of the results.**
The sentence line 304 « The results presented in the next sections of this study were obtained with the configuration 0. » was replaced with « The next sections present firstly flow intermittence modelling results obtained with configuration~0 in Sections 3.3 and 3.4 and secondly the uncertainty related to the input data (size of the training sample with configuration~1, type of flow state observed data, and geology data) in Section 3.5. »

The following sentences were added Line 357:
« The importance of the covariates obtained with configuration 1 is very close to that obtained with configuration 0 (Fig.10) which shows that for this study the importance of the covariates is not very sensitive to the size of the training sample (see more details in Supplementary Material). »

**19. Figure 11. The name of the Finnish river needs to be corrected.**
The name of the Finnish catchment was also corrected in Figures 5, 7, 9, 12, et 13

**20. Lines 356 to 260. It is not clear if this paragraph only refers to the Albarine catchment. Please clarify.**
The title of subsection 3.5.3 was changed to « Sensitivity to the geology data in the Albarine DRN » (Line 385).

**21. Line 382: Should it be "are not-known"?**
Line 412 was corrected to « The use of the RF model enables to simulate flow intermittence even if the processes causing the drying up are not known or understood precisely beforehand since it does not require a representation of physical processes, but links covariates to observed states of flow. »

**22. Line 439. You could support this claim using previous expert elicitation studies that looked into expert perception uncertainty. For example: https://doi.org/10.5194/hess-26-5605-2022 or https://doi.org/10.1002/2015WR018461**
The following sentences were added lines 480-484:
« Expert elicitation in hydrology has already shown benefits, particularly when tangible data are missing (Ye et al, 2008, Warmink et al. 2011, Sebok et al. 2016, Sebok et al. 2022). These studies do show differences in the individual perceptions of the experts consulted, but by consulting a larger number of experts (in this study, only 1 or 2 experts were consulted per studied DRN) and by applying protocols similar to the ones proposed in these studies, the uncertainty linked to individual perception could be reduced, or at least quantified. »

**23. Lines 479 to 483. As you state that you will use this approach to assess impacts of climate change, it might good to add a line about the implications of using it for such kind of studies. For instance, mentioning a bit on what you wrote in line 396.**
The following lines were added Line 529 :
« One of the challenges will therefore be to analyse the hybrid model's ability to extrapolate the flow state of river sections in a future climate. In particular, it will be necessary to analyse the model's ability to simulate changes in flow regime (for example, the transition from perennial to intermittent flow) outside its training period. »

**24. Line 265. You repeated "which shows the" in the sentence. Please remove one.**
Line 286 was corrected : « However, the flow state predicted by the RF model is in good agreement with the observed flow states, which shows the usefulness of the coupling between the spatialized hydrological JAMS-J2000 model and the RF model. »